

**Large Scale Physical Modelling Study of a Flexible Barrier under the**
**Impact of Granular Flows**
by
**Dao-Yuan TAN**
Department of Civil and Environmental Engineering
The Hong Kong Polytechnic University, Hung Hom, Kowloon, Hong Kong, China
Email: t.daoyuan@connect.polyu.hk
**Jian-Hua YIN** (Chair Professor and Corresponding Author)
Department of Civil and Environmental Engineering
The Hong Kong Polytechnic University, Hung Hom, Kowloon, Hong Kong, China
Tel: (852) 2766-6065, Fax: (852) 2334-6389, Email: cejhyin@polyu.edu.hk
**Wei-Qiang FENG**
Department of Civil and Environmental Engineering,
The Hong Kong Polytechnic University, Hung Hom, Kowloon, Hong Kong. China
Email: fengweiqiang2015@gmail.com
**Jie-Qiong QIN**
Department of Civil and Environmental Engineering
The Hong Kong Polytechnic University, Hung Hom, Kowloon, Hong Kong, China
Email: jieqiong.qin@connect.polyu.hk
And
**Zhuo-Hui ZHU**
Department of Civil and Environmental Engineering
The Hong Kong Polytechnic University, Hung Hom, Kowloon, Hong Kong, China
Email: zhuo-hui.zhu@connect.polyu.hk
Manuscript submitted to *Natural Hazards and Earth System Sciences* for possible
publication as a Technical Paper
June 2018





**Abstract:**
Flexible barriers are being increasingly applied to mitigate the danger of debris flows.
However, how barriers can be better designed to withstand the impact loads of debris
flows is still an open question in natural hazard engineering. Here we report an
improved large-scale physical modelling device and the results of two consecutive
large-scale granular flow tests using this device to study how flexible barriers react
under impact from granular flows. In the study, the impact force directly on the flexible
barrier and the impact force transferred to the supporting structures are measured,
calculated and compared. Based on the comparison, the impact loading attenuated by
the flexible barrier is quantified. The hydro-dynamic and hydro-static approaches are
also validated using the calculated impact forces.
**KEYWORDS:** Large-scale tests; granular flow; flexible barrier; impact loading



## 1. Introduction

Debris flows, as one of the most disastrous natural geohazards, have caused destructive damage to human lives and their habitations in many countries such as USA, Japan, and China (Takahashi 2014; Hungr 1995; Ishikawa *et al.* 2008; Su *et al.* 2017). In a mountainous area where a large amount of loose sediment is present, multiple debris flows can occur under intensive heavy rains (Xu *et al.* 2012; Yagi *et al.* 2009; Chen *et al.* 2017). Protective systems such as concrete check dams are usually installed in areas threatened by debris flows to prevent the damage (Santi *et al.* 2011). Nowadays, researchers have found that flexible barriers, which were firstly used in rockfall prevention, are effective to trap debris flows (Canelli *et al.* 2012; Wendeler *et al.* 2007; Cui *et al.* 2015; Hu *et al.* 2006; Kwan *et al.* 2014). Compared to conventional rigid concrete check dams, flexible barriers have a few obvious advantages: economical, efficient in impact energy absorption, easy to be installed and adaptable to various terrains (Ashwood and Hungr 2016; Wendeler and Volkwein 2015). However, the performance of a flexible barrier subjected to the impact of debris flows has not been fully understood. The efficiency of loading reduction by flexible barriers has not been quantified yet. Therefore, further research on the interaction between debris flows and a flexible barrier is urgently required.

Physical modelling has been widely used in geotechnical engineering research because of its excellent controllability in testing conditions and good reliability of testing results (Paik *et al.* 2012; Wendeler *et al.* 2006; Bugnion *et al.* 2012; DeNatale *et al.* 1999). Scaling is a key parameter in experiment design for studying debris flows because it can affect the interaction between particles in a granular flow. In miniaturized debris flows generated in small-scale tests, the effects of viscous shear resistance, friction, and



cohesion are over-represented, whereas the effects of excess pore-fluid pressure, which
are generated by debris dilation or contraction, are under-represented (Iverson 2015).
Considering the scale effects, some researchers use large-scale physical models or field-
scale experimental sites to study debris flows (DeNatale *et al.* 1999; Paik *et al.* 2012;
Bugnion *et al.* 2012; Iverson 2015). WSL (2010) conducted a series of full-scale tests
to study the interaction between multiple debris flows and a prototype flexible barrier.
Large-scale physical modelling tests are also selected by the authors to investigate the
interaction between a flexible barrier and dry granular flows.

A typical flexible barrier usually consists of two main components: a flexible ring net
and supporting structures (supporting posts holding the ring net, strand cables and
foundations supporting the posts). The impact loading from a debris flow is firstly
attenuated by the flexible ring net with large deformation, then transfers to the cross-
tension cables, which form the outline frame and stretch the ring net, and finally to the
posts and the supporting cables. Generally, break elements are installed on the
supporting cables to reduce load peaks transferred to the foundations (Volkwein 2014).
In this study, break elements are replaced by large capacity tension link transducers to
measure the impact loading transferred to the supporting structures.

Impact loading estimation is key to the design of a flexible barrier for debris flow
mitigation (Volkwein *et al.* 2011). Simple approaches are commonly used by designers
in impact loading estimation because they require only a few parameters in the
calculation. There are two widely accepted simple approaches: the hydro-dynamic
approach and the hydro-static approach. The hydro-dynamic approach is based on
momentum conservation. In this approach, the impact period is taking as an ideal flow



with a uniform velocity impacting the barrier and deviating along the vertical direction.
The impact loading is calculated from the momentum change of the decelerated debris
flow during the impact (Hungr *et al.* 1984; Armanini 1997). The hydro-static approach,
on the other hand, is calculated from the earth pressure of deposited debris (Kwan and
Cheung 2012). Both approaches adopt empirical coefficients to reach a good accuracy
in predicting real cases.

The estimation of impact force with the hydro-dynamic approach (Hungr *et al.* 1984)
is expressed as follows:
$$F_{calculated} = \alpha \rho_{bulk} v_0^2 h w \qquad (1)$$
where $\rho_{bulk}$ is the bulk density of a debris flow, $v_0$ is the velocity of the debris flow, $h$ is
the height of the debris flow, $w$ is the width of the debris flow, which is normally
represented by the width of the flowing channel, and $\alpha$ is the dynamic coefficient.
Hungr *et al.* (1984) proposed a value of 1.5. Kwan and Cheung (2012) suggested a
value of 2.0 considering the flexibility of flexible barriers. A range between 1.5 and 5
was given by Canelli et al (2012).

The hydro-static approach (Lichtenhahn 1973; Armanini 1997) is given as follows:
$$F_{calculated} = \kappa \rho_{bulk} g h_{deposit}^2 w \qquad (2)$$
where $\kappa$ is the static coefficient, which is suggested as 1.0 in the calculation (Kwan and
Cheung 2012). $g$ is gravitational acceleration, and $h_{deposit}$ is the deposition height of the
debris flow.

This paper aims to study the motions of multiple granular flows and the performance
of a flexible barrier under the impact of granular flows. The data from well-arranged



transducers and high-speed cameras in the debris flow impact tests are presented and
analyzed in this paper. The motions of two consecutive granular flows are described.
The impact forces on the flexible ring net and the supporting structures of the flexible
barrier are calculated respectively. Using the calculated results, the contribution of
flexibility to impact loading reduction is quantified, and simple approaches for impact
force estimation are verified.

**2. Experiment setup and instrumentation**
*2.1 Description of the experiment apparatus*
A large-scale testing device is built in the Road Research Lab of the Hong Kong
Polytechnic University with a length of 9.5 m, a height of 8.3 m and a width of 2 m.
The view of the experiment setup is plotted in Fig.1. This facility can be divided into 4
main components: (i) a reservoir with the capacity of 5 m$^3$ at the top of the device, (ii)
a novel quick flip-up door opening system at the front vent of the reservoir, (iii) a
flexible barrier with supporting posts and cables, and (iv) a flume linking the reservoir
and the flexible barrier. The prototype flexible barrier with a width of 2.48 m is made
up of steel rings with a diameter of 300 mm (No. ROCCO 7/3/300, Geobrugg), which
are commonly used in rockfall mitigation in European and Hong Kong. This ring net is
covered by a flexible secondary wire net with the mesh size of 50mm to provide a high
trapping rate for the granular flows. Two parallel posts that can rotate in the plane of
impact are installed to stretch and support the ring net, and each post is supported by
two inclined strand cables. The flume has a length of 7 m, an inner width of 1.5 m and
an inclination angle of 35 °. Side walls of the flume are made up of tempered glass to
provide a clear observation of generated granular flows and their interactions with the
flexible barrier.




## 2.2 Instrumentation

To monitor the performance of a flexible barrier under the impact of granular flows,
this device is instrumented with a well-arranged high-frequency measurement system.
Two types of transducers are installed on the flexible protection system: mini tension
link transducers and high capacity tension link transducers. The mini tension link
transducers were calibrated in the soil laboratory with a maximum loading of 20 kN.
The calibration is plotted in Fig.2. Those transducers are installed on the flexible ring
net to measure the impact force on the flexible ring net directly. Specifically, the central
area of the flexible ring net, which consists of 5 connected rings, is separated from the
main net and reconnected to the neighboring rings by 10 mini tension link transducers.
Fig.3 presents the measured central area and the arrangement of all the mini tension
link transducers on the flexible ring net. The high capacity tension link transducers with
a certified capacity of 50 kN are installed on the supporting cables of the posts (seen
Fig.1 (b)). A data-logger with the capability of sampling 48 transducers at 1000 Hz
simultaneously is used to collect the data of all transducers. Two high-speed cameras
capable of capturing a resolution of 1024 ×768 pixels at a sampling rate of 1000 frames
per second are used to capture the motions of the granular flows and the deformation
of the flexible barrier under impact. One high-speed camera is located at the right side
of the barrier, and the other one is set in front of the barrier.

## 2.3 Experiment material and procedures

The sample of material used in the tests is plotted in Fig.4, and their properties are listed
in Table 1. Two consecutive tests, named Test 1 and Test 2 were conducted using the



same granular material. In test 1, the granular flow travelled on the steel plate of the
flume and impacted an empty flexible barrier. While in Test 2, the granular flow moved
on the upper surface of the deposition in Test 1 to simulate the second surge in multiple
flows. At the beginning of tests, the door was flipped up in less than 0.5 s with the help
of a novel door opening system to generate a uniform granular flow. The datalogger
started to obtain data several seconds before the triggering of the granular flow to obtain
initial values of all the transducers. Simultaneously, the high-speed cameras started to
capture the motion of the granular flow and its interaction with the flexible barrier
during the impact.

**3.  Test results**
*3.1 Motion and impact of granular flow in Test 1*
In test 1, the initial time of the impact has been readjusted to 0 s in all plotted data and
selected video frames, and the negative value of time represents the moment before the
interaction. By tracking the motion of the granular flow with high-speed cameras, the
speed of the granular flow was 5 m/s, which was relatively low compared with the
measured velocities from 2 m/s to 12 m/s in literatures (Arattano and Marchi 2005;
Prochaska *et al.* 2008; Berti *et al.* 1999). The deposition height of the granular flow,
the maximum horizontal deformation of the flexible barrier and the tensile force of
Transducer 1 with time are plotted in Fig.5. It can be seen that the deposition height of
trapped aggregates rises almost linearly with time and reaches 0.55 m at the time of 1.0
s, and the horizontal deformation of the barrier increases from an initial value of 0.262
m to 0.481 m at the time of 1.0 s. The side profiles of the deposited aggregates at
different times are plotted in Fig.6. From 0 s to 1.0 s, the front portion of the granular



flow shot up, impacted the barrier directly and deposited as a wedge-shaped dead zone
at the base of the flexible barrier. The following granular flow climbed on the top
surface of the previous stationary deposition, impacted the flexible barrier, and
deposited behind the barrier layer by layer. After 1.0 s, the following granular front
deposited behind the deposition wedge. It is worth noting that the tensile force on the
net keeps increasing even the deposition height of the granular flow reach the maximum
value (see Fig.5), and this phenomenon indicates that the granular flow can
continuously exert impact pressure on the flexible barrier via the deposition wedge.

*3.2 Impact loading analysis in Test 1*
Tensile forces recorded by the mini tension link transducers between rings are plotted
in Fig.7. Signals of the transducers have some noises due to the intensive impacts from
thousands of aggregates during the impact period. Thus, trend lines are added into those
figures to clarify the changes of tensile forces. It can be observed that a gradual rise of
static load and two dynamic impact peaks in the signals of most transducers. The first
impact peak occurred at the beginning of the impact, and the second impact peak
appeared at the end of the impact. These two peaks are much smaller than the
accumulated static load. It is indicated that the dynamic load and the static load co-
existed in the impact, and the static load was dominant. Besides, transducers connected
to the bottom cross-tension cable (Transducer 7 and Transducer 8) present negative
values, which shows that they were compressed in the impact. Fig.8 presents typical
frames recorded by the side-view camera and the front-view camera combined with the
signal from Transducer 1. From this figure, it can be indicated that the first dynamic
impact peak came from the direct impact of the first debris front on the flexible barrier,



and the gradual increase of the static load was caused by the deposition of the
aggregates. With the growth of the deposition zone, the impact loading of the following
granular flow was finally fully resisted by the deposition cushion. Afterward, only static
earth pressure of the deposition acted on the flexible barrier.

*3.3 Motion of granular flow in Test 2*
The second granular flow was triggered after Test 1 to simulate the second flow in a
multiple debris flow event. In Test 2, the granular flow travelled on the top surface of
the deposition in Test 1 and came to rest without reaching the net. The motion of the
granular flow in Test 2 is plotted in Fig.9. In that figure, the initiated time of the granular
flow is readjusted to 0 s. It can be found that the granular flow had a thick front when
it was firstly triggered, then the thickness kept decreasing during movement. Based on
the recording of the side-view camera, the side-view of depositions in the two tests and
the velocity change with the flowing distance of the granular flow in Test 2 are plotted
in Fig.10. Thickness and velocity of the front reduced dramatically with the increase of
the moving distance and finally stopped at 0.7 m before the flexible barrier.
Correspondingly, no signal fluctuation and deformation increment of the flexible
barrier were recorded by the transducers and the high-speed cameras. The reason for
the flow stopping before the flexible barrier is the large basal friction from the rough
interface between the moving granular flow and the deposition and the low fluidity of
the dry granular flow. The multi-flow tests show that the impact from the latter arrived
debris flows can be attenuated or eliminated by the resistance from the deposition of
the previous debris flow in a multiple debris flow event.



## 4. Data analysis

### *4.1 Direct measurement of the impact force on the flexible barrier*

As mentioned above, the central area is separated from the main ring net and reconnected to neighboring net rings by mini tension link transducers. Two assumptions are made to simplify the measurement of the impact loading on a flexible ring net. The deformation of the ring net is assumed similar to a membrane, and the deformation in the measured area is assumed cone symmetric. Based on the assumptions, the loading situation in the cross-section of the measured area which contains Transducer $i$ and Transducer $i+1$ is analyzed and shown in Fig.11. Thus, the impact force on the cross-section can be calculated with the following equation:

$$F_{impact,i,i+1} = F_{tensile,i} \cdot \cos\frac{\theta}{2} + F_{tensile,i+1} \cdot \cos\frac{\theta}{2} \qquad (3)$$

where $F_{tensile,i}$ and $F_{tensile,i+1}$ are the maximum tensile forces on Transducer $i$ and Transducer $i+1$ installed in the measured area, $\theta$ is the included angle between the opposite transducers, $F_{impact,i,i+1}$ is the calculated impact force on this cross-section. Since the deformation in the measured area is assumed cone symmetric, $\theta$ is a constant in all cross-sections formed by two opposite transducers. Thus, for the measured area with $n$ transducers, the maximum impact force, $F_{measured}$, can be calculated with the following equation:

$$F_{measured} = \cos\frac{\theta}{2} \cdot \sum_{i=1}^{i=n} F_{tensile,i} \qquad (4)$$

In our study, the maximum tensile forces on all the transducers are measured and plotted in Fig.12, and $\theta$ can be measured from the photograph taken at the moment of the largest deformation as shown in Fig.13.




The impact pressure from the granular flow is assumed to be uniformly distributed in
the cross-section area of the flume width multiplied by the height of the debris
deposition, which covers the measured central area. Combined with Eq. 4, the following
equation is given to calculate the distributed impact loading on a flexible ring net as:
$$F_{impact} = F_{measured} \cdot \frac{A_{impact}}{A_{measured}} = \cos\frac{\theta}{2} \cdot \sum_{i=1}^{i=n} F_{tensile,i} \cdot \frac{A_{impact}}{A_{measured}}$$
(5)

where $A_{impact}$ and $A_{measured}$ represent the actual impact cross-section area and the
measured central area in the test as shown in Fig.10. All the parameters and calculated
results are listed in Table 2.

*4.2 Calculation of Loading Reduction Rate (LRR)*
The flexible ring net is supported by two posts that can rotate in the plane of the flow
direction, and each post is supported by two inclined steel strand cables. Therefore, the
impact force transferred from the flexible barrier to the supporting posts can be
calculated from the tensile forces carried by the supporting cables in the direction of
impact. Based on the symmetrical arrangement of the cables and the posts with respect
to the flexible barrier, as plotted in Fig.14 (a), the loading situations of the posts and
the supporting cables located on both sides of the flexible barrier are also symmetrical
when they are under a uniform impact pressure. Thus, the left post and its supporting
cables: Cable A Left and Cable B Left are selected as the analysis objects. The force
analysis of the supporting cables is divided into two steps:
Firstly, forces on Cable A Left and Cable B Left are decomposed into components in
the rotation plane of the post based on the top-view sketch (seen Fig.14(a)):





$$F_{AL,H} = F_{AL} \cdot \cos \alpha \tag{6}$$

$$F_{BL,H} = F_{BL} \cdot \cos \beta \tag{7}$$

where $F_{AL}$ and $F_{BL}$ are the measured maximum tensile forces on Cable A Left and Cable B Left during the impact, $F_{AL,H}$ and $F_{BL,H}$ are the components of $F_{AL}$ and $F_{BL}$ decomposed in the rotation plane of the left post, and $\alpha$, $\beta$ are the included angles between Cable A, Cable B and the rotation plane of the post.

Secondly, based on the calculated $F_{AL,H}$ and $F_{BL,H}$, components of the tensile forces on Cable A Left and Cable B Left in the direction of impact can be calculated based on the left-side-view sketch (seen Fig.14 (b)):

$$F_{AL,imapct} = F_{AL,H} \cdot \cos \gamma \tag{8}$$

$$F_{BL,imapct} = F_{BL,H} \cdot \cos \delta \tag{9}$$

where $F_{AL,impact}$ and $F_{BL,impact}$ are the components of tensile forces on Cable A Left and Cable B Left in the direction of impact, and $\gamma$, $\delta$ are the included angles between Cable A, Cable B and the direction of impact.

It is defined that the direction of the supporting force, which is opposite to the direction of the impact force, is the positive direction. Thus, the components of the tensile forces on the left cables in the direction of impact ($F_L$) can be calculated by substituting Eqs. (6) and (7) into Eqs. (8) and (9):

$$\begin{aligned} F_L &= F_{BL,imapct} - F_{AL,imapct} = F_{BL,H} \cdot \cos \delta - F_{AL,H} \cdot \cos \gamma \\ &= F_{BL} \cdot \cos \delta \cdot \cos \beta - F_{AL} \cdot \cos \gamma \cdot \cos \alpha \end{aligned} \tag{10}$$



Finally, based on the conservation of angular momentum and the symmetrical
arrangement of the cables and the posts with respect to the flexible barrier, the
equivalent impact force can be calculated from the tensile forces on the supporting
cables with the following equation:
$$F_{Cables,equivalent} = \frac{l_{post}}{l_{impact}} \left[ (F_{BL} + F_{BR}) \cdot \cos\delta \cdot \cos\beta - (F_{AL} + F_{AR}) \cdot \cos\gamma \cdot \cos\alpha \right] \qquad (11)$$
where $F_{Cables,equivalent}$ is the equivalent impact force calculated from the tensile forces on
the supporting cables, $l_{post}$ is the distance between the rotation fulcrum of the post and
the connecting point of the cables, $l_{impact}$ is the distance between the rotation fulcrum of
the post and the equivalent impact height of the granular flow. $F_{AL}$, $F_{AR}$, $F_{BL}$, and $F_{BR}$
are the measured maximum tensile forces on the supporting cables. Their values are
presented in Fig.13. All parameters, as well as the calculated results, are listed in Table

327    2.


It is found that flexibility of flexible barriers makes an obvious contribution to the
reduction of the impact loading from a debris flow (Volkwein 2014; Song *et al.* 2017).
Since almost all the debris material was trapped in this study, the load reduction mainly
attributes to the large deformation of the flexible ring net during the impact. To quantify
the contribution of flexibility to impact loading reduction, the Loading Reduction Rate
(LRR) of the flexible barrier is defined as:
$$LRR = \frac{F_{impact} - F_{Cables,equivalent}}{F_{impact}} \cdot 100\% \qquad (12)$$
LRR in the granular flow tests is calculated and presented in Table 2. It is found that
around 28 % of the impact loading from the dry granular flow in Test 1 was attenuated

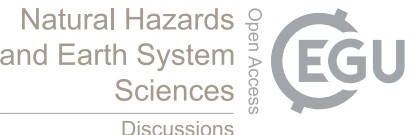

by the flexible barrier.

### 4.3 Comparison of simple approaches with measured impact forces

Two widely accepted simple approaches for impact force estimation: hydro-dynamic
approach and hydro-static approach (Kwan and Cheung 2012; Volkwein 2014; Song *et*
*al.* 2017; Ashwood and Hungr 2016) are compared in this section to validate their
applications in the design of flexible barriers. The parameters and the measured impact
forces on different components in Test 1 are used in this comparison (see Table 3). To
quantify the accuracies of the simple approaches, Relative Error (RE) is defined as:
$$RE = \left| \frac{F_{calculated} - F_{measured}}{F_{measured}} \right| \times 100\% \qquad (13)$$

As listed in Table 3, it can be found that the hydro-dynamic approach with the dynamic
coefficient of 2.0 has the best performance in estimating the impact force on the flexible
net with a small deviation of 5.8 %. While the hydro-static approach with the static
coefficient of 1.0 fits quite well with the measured impact force on the supporting
structures. This is reasonable since the dynamic impact from the granular flow can be
attenuated by the flexible ring net, and the static loading can be transferred to the
supporting structures. This phenomenon is also proved by the gradually increased
tensile forces on Cable B Left and Cable B Right shown in Fig.13 (b). Thus, in the
design of a flexible barrier for debris flow mitigation, the hydro-dynamic approach and
the hydro-static approach can be used in the design and the selection of the flexible ring
net and the supporting structures, respectively. Even the dynamic coefficient and the
static coefficient suggested by Kwan and Cheung (2012) are feasible in this study, more
tests are required to further verify more appropriate coefficients before they can be used
in the design.




**5. Conclusions**

In this paper, an improved large-scale physical modelling facility for debris flow
research and a well-arranged high-frequency measurement system are introduced.
Using this device, two tests were performed to study the behavior of a flexible barrier
subjected to the impacts of granular flows. From the experimental data and their
analysis, key findings and conclusions are summarized and presented as below:
(a) In Test 1, the front of the granular flow impacted the flexible ring net directly,
deposited behind the barrier layer by layer, and formed a deposition wedge. After
1.0 s, the following granular flow deposited behind the deposition wedge.
(b) The static loading and the dynamic loading co-existed in the impact process, and
the static loading was dominant. The static loading attributed to the gradual
deposition of aggregates, and the dynamic loading was caused by the impact of the
granular front. The latter arrived granular front applied impact loading on the
flexible barrier via the deposition wedge. With the deposition of aggregates, the
stationary debris formed a cushion behind the barrier and attenuated all the impact
loading from the following granular front.
(c) In Test 2, the second granular flow in a multiple flow event was performed. The
velocity and the flow depth of the granular flow decreased during movement, and
the front stopped before it can reach the flexible barrier due to the large basal
friction between the moving granular flow and the granular deposition and the poor
fluidity of the dry granular flow.
(d) The impact loading on a flexible ring net was directly measured from the tensile
forces on the central area of the flexible ring net. In Test 1, the measured impact

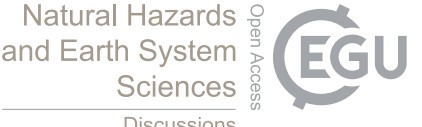

force on the flexible ring net is 10.96 kN.
(e)  The contribution of flexibility to impact loading reduction is quantified by

introducing the Loading Reduction Rate (LRR). By calculating the impact loading

transferred to the supporting structures, it can be concluded that almost 28 % of the

impact loading from the granular flow was attenuated by the flexible ring net in

Test 1.

(f)  From the comparisons of the hydro-dynamic approach and the hydro-static

approach with the measured impact forces on different components, it is found that

the hydro-dynamic approach with the dynamic coefficient of 2.0 fits well with the

measured impact force on the flexible ring net, and the hydro-static approach with

the static coefficient of 1.0 has a good performance in estimating the impact force

on the supporting structures.


The motion characteristics of the multiple granular flows indicate that the motion and
the impact of the following debris flow can be resisted or eliminated by the deposition
of previous debris flow. By applying the LRR and suitable impact loading estimation
approaches, the design of a flexible barrier can be optimized by designing different
components such as the flexible ring net and the supporting structures individually,
which provides a safer and more economical method in design. In the future, the tests
of rapid debris flows will be conducted to investigate the behavior of debris flows and
examine the performance of a flexible barrier under the impact of rapid debris flows.

**Acknowledgement**
The authors acknowledge the financial support from Research Institute for Sustainable
Urban Development of The Hong Kong Polytechnic University (PolyU). The work in





this paper is also supported by a National State Key Project "973" grant (Grant No.:
2014CB047000) (sub-project No. 2014CB047001) from Ministry of Science and
Technology of the People's Republic of China, a CRF project (Grant No.:
PolyU12/CRF/13E) from Research Grants Council (RGC) of Hong Kong Special
Administrative Region Government of China. The financial supports from PolyU
grants (1-ZVCR. 1-ZVEH. 4-BCAU, 4-BCAW, 4-BCB1, 5-ZDAF) are acknowledged.
This paper is also supported by Research Centre for Urban Hazards Mitigation of
Faculty of Construction and Environment of PolyU.

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



# Tables



**Table 1.** Main properties of aggregates used in the test

| Main properties | Values |
|---|---|
| *The total volume of aggregates in Test 1 and Test 2 ($m^3$)* | 4 |
| *Particle diameters (mm)* | 15 ~ 30 |
| *Internal friction angle (°)* | 36 |
| *Interface friction angle (°)* *(between aggregates and painted steel plate)* | 28 |
| *Bulk density (kg/$m^3$)* | 1600 |




**Table 2.** Values of measured parameters and calculated results in Test 1

| Parameters and results | Values |
|---|---|
| *Moving speed (m/s)* | 5 |
| *Included angle θ (°)* | 130 |
| $A_{measured}$ ($m^2$) | 0.644 |
| $A_{impact}$ ($m^2$) | 1.44 |
| $\sum\limits_{i=1}^{i=n} F_{tensile,i}$ (kN) | 11.59 |
| $F_{measured}$ (kN) | 4.9 |
| $l_{impact}$ (m) | 0.242 |
| $l_{post}$ (m) | 2.7 |
| $h_{debirs}$ (m) | 0.086 |
| $h_{deposit}$ (m) | 0.58 |
| α (°) | 62 |
| β (°) | 24 |
| γ (°) | 76 |
| δ (°) | 60 |
| $F_{AL}$ (kN) | 0.062 |
| $F_{AR}$ (kN) | 0.062 |
| $F_{BL}$ (kN) | 0.79 |
| $F_{BR}$ (kN) | 0.79 |
| $F_{Cables,equivalent}$ (kN) | 7.89 |
| $F_{impact}$ (kN) | 10.96 |
| *Loading Reduction Rate (LRR) (%)* | 28.01 |







**Table 3.** Comparisons of the calculated impact forces using simple approaches with the measured impact forces on different components of a flexible barrier in Test 1

| Simple approaches for impact force estimation | Calculated impact force (kN) | RE with impact force on the flexible net (%) $F_{impact}=10.96\ kN$ | RE with impact force on the supporting structures (%) $F_{Cables,equivalent} =7.89$ kN |
|---|---|---|---|
| $F_{calculated} = \alpha\rho_{bulk}v_0^2hw$ (hydro-dynamic approach with α=1.5) (Hungr et al. 1984) | 7.74 | 29.4 | 1.9 |
| $F_{calculated} = \alpha\rho_{bulk}v_0^2hw$ (hydro-dynamic approach with α=2) (Kwan and Cheung 2012) | 10.32 | **5.8** | 30 |
| $F_{calculated} = \kappa\rho_{bulk}gh_{deposit}^2w$ (hydro-static approach with κ=1) (Kwan and Cheung 2012) | 7.92 | 27.7 | **0.38** |





**Figure lists**


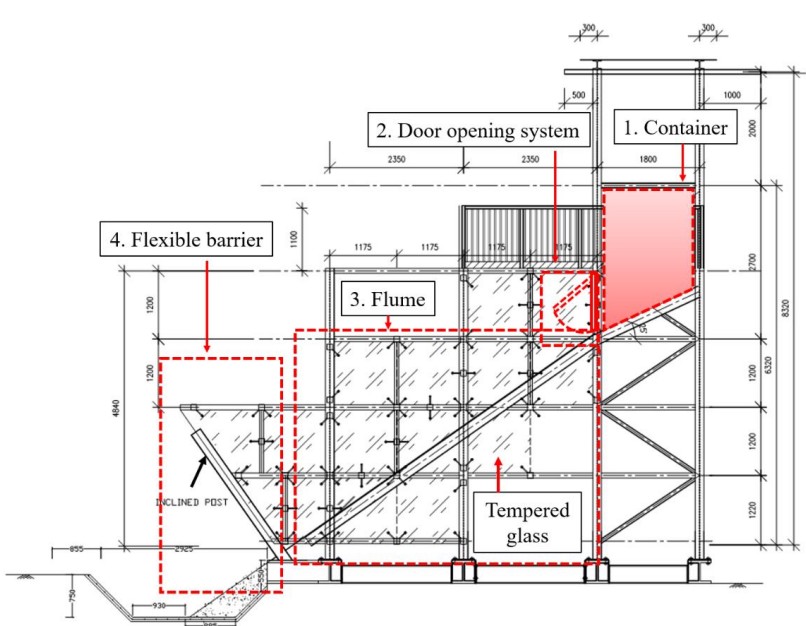


(a)

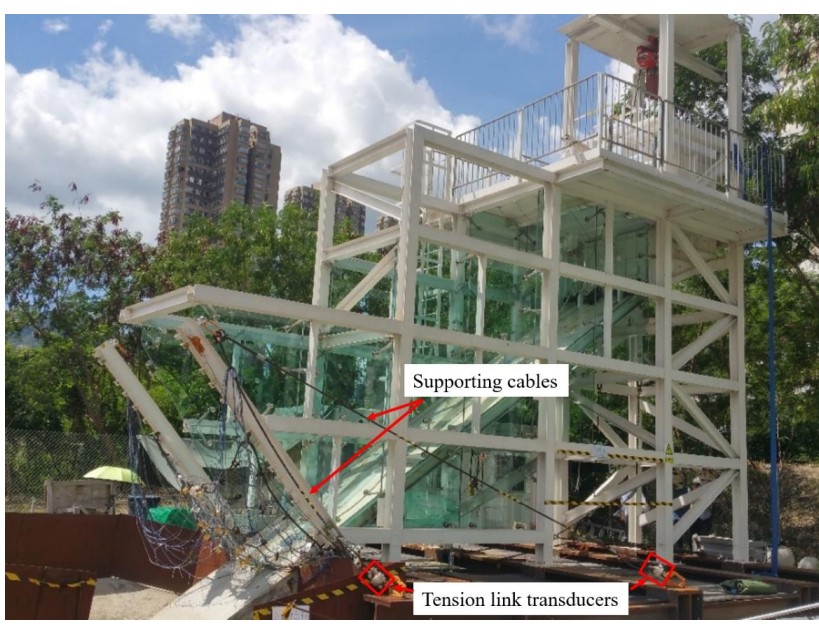


(b)

**Fig.1.** (a) side view of a large-scale physical model design (unit in mm) and (b) view
of the physical modelling facility constructed at a site in Hong Kong





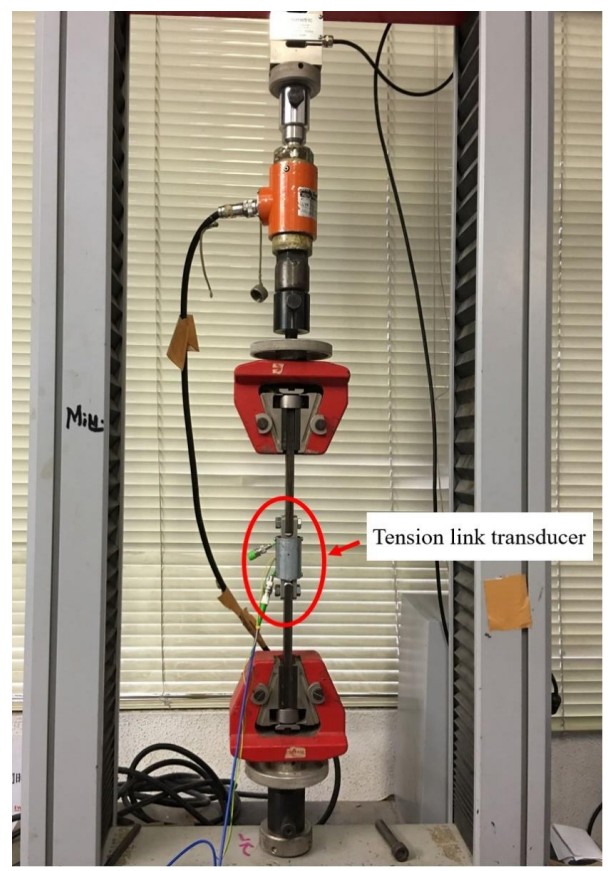


**Fig.2.** Calibration of a tension link transducer





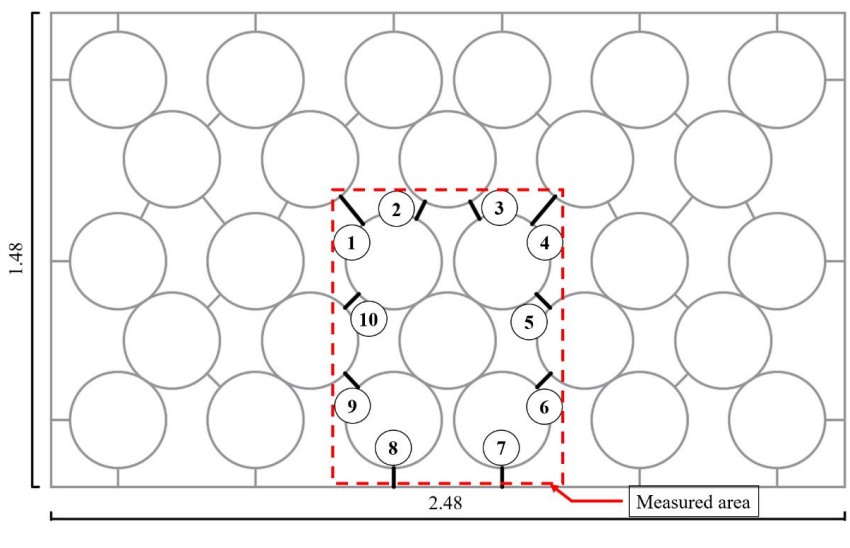


(a)

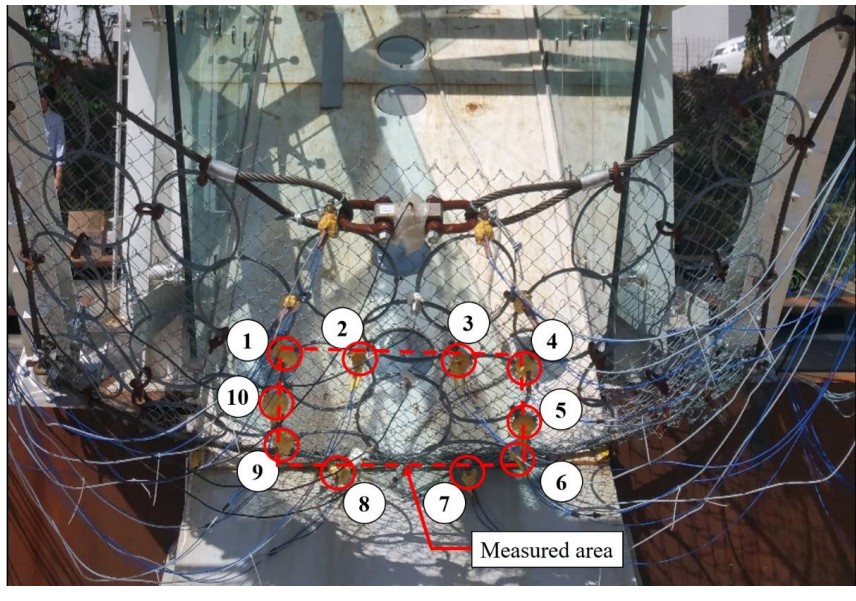


(b)

**Fig.3.** (a) schematic diagram of a flexible barrier and (b) front view of the flexible
barrier with numbered tension link transducers between rings and the
measured area in the physical model (unit in m)




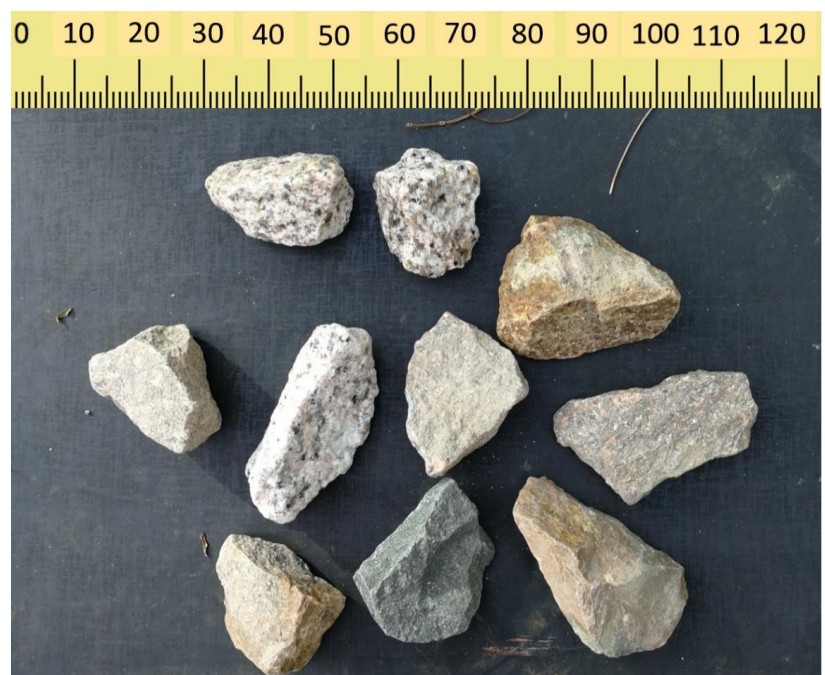



**Fig.4.** Aggregate samples in the granular flow impact tests (unit in mm)




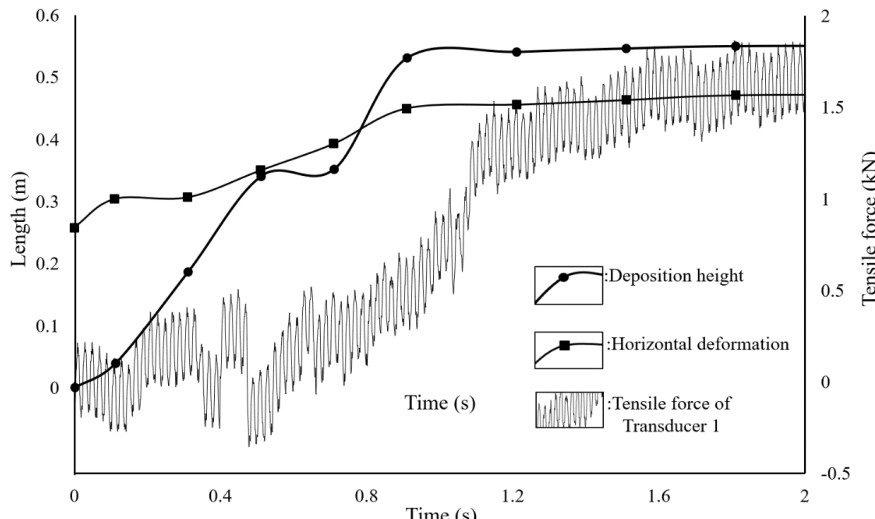


**Fig.5.** Relation of the deposition height of the granular flow, horizontal deformation
of the flexible barrier and tensile force of Transducer 1 with time in Test 1






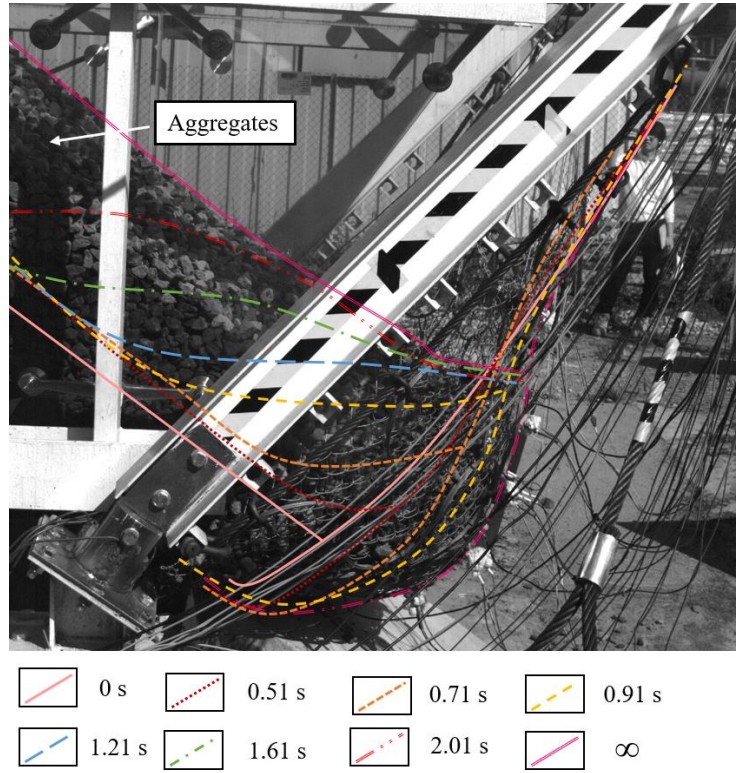


**Fig.6.** Side profiles of deposited aggregates at different times in Test 1


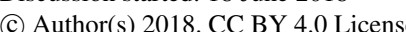


**Fig.7.** Recorded forces with time by the mini tension link transducers between rings
in Test 1




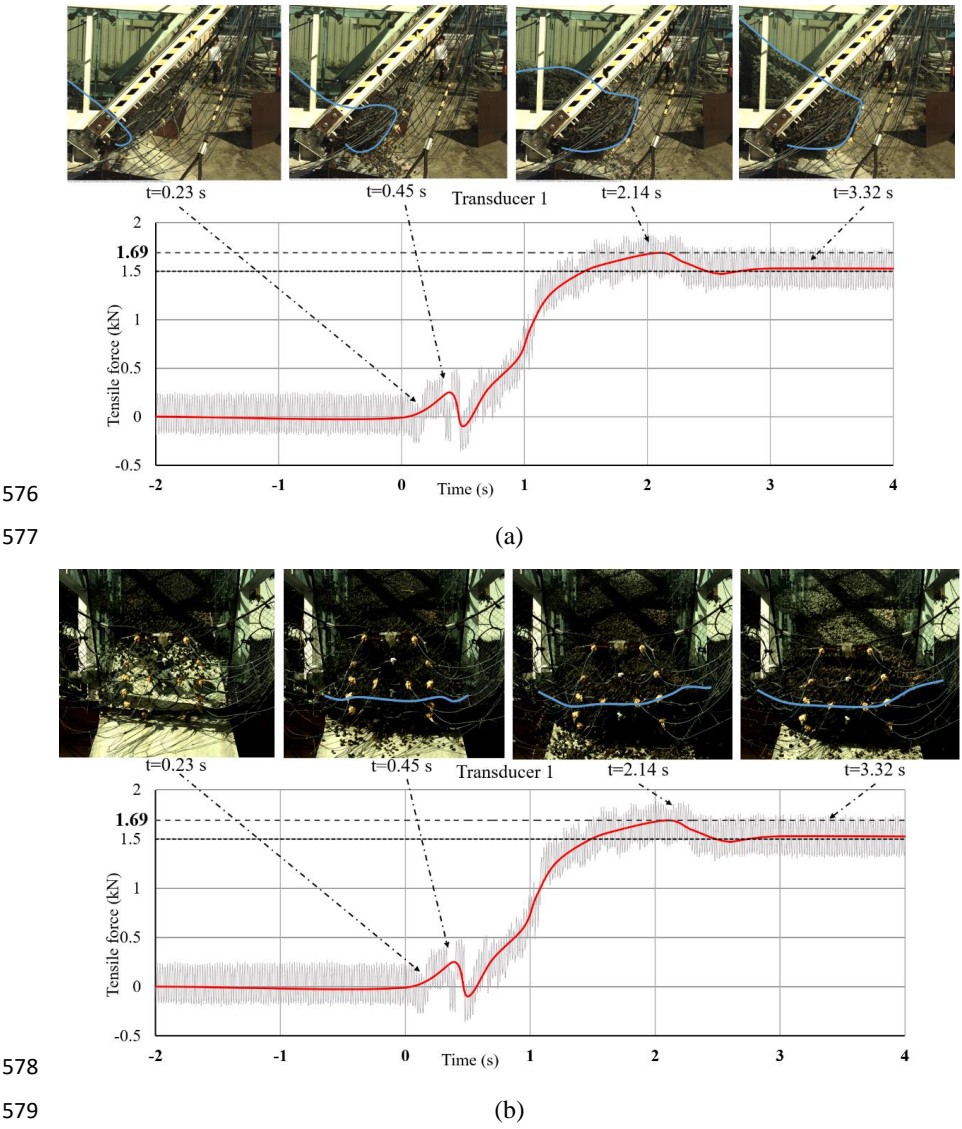


(a)


(b)

**Fig.8.** Interpretation of the typical video frames in Test 1 recorded by (a) the side-
view camera and (b) the front-view camera with the data of tensile force from
Transducer 1





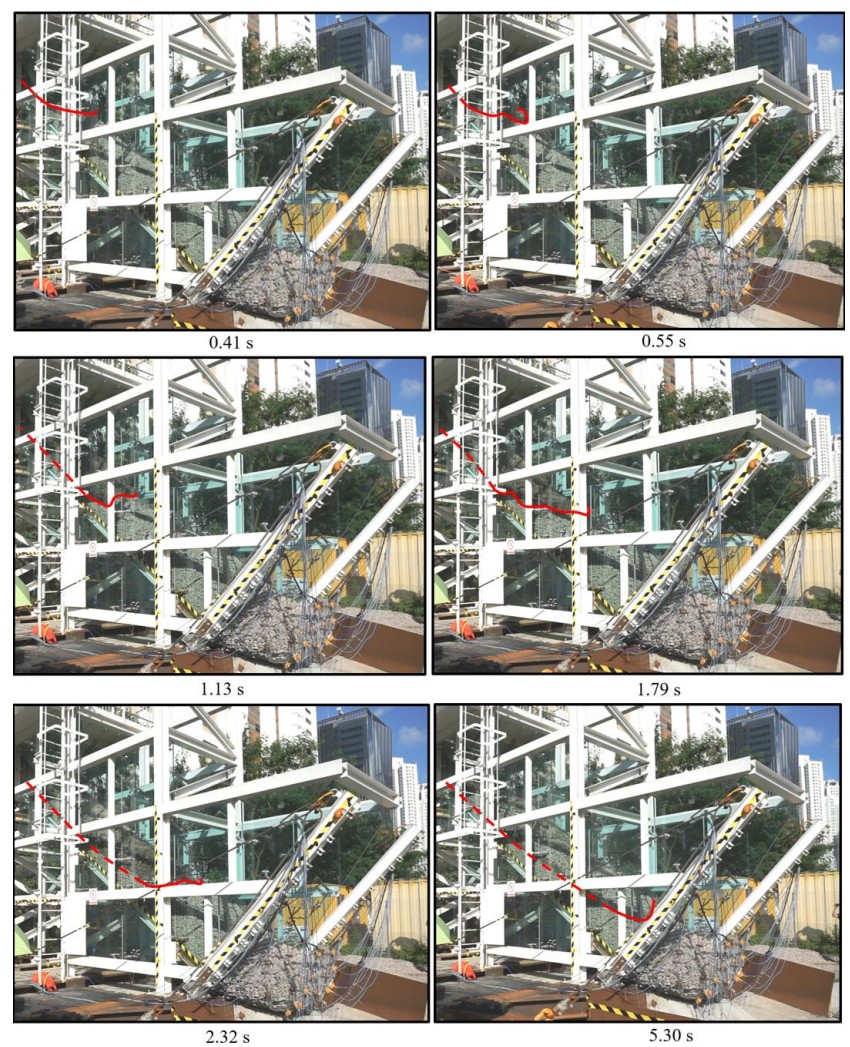


**Fig.9.** Motion of the granular flow in Test 2






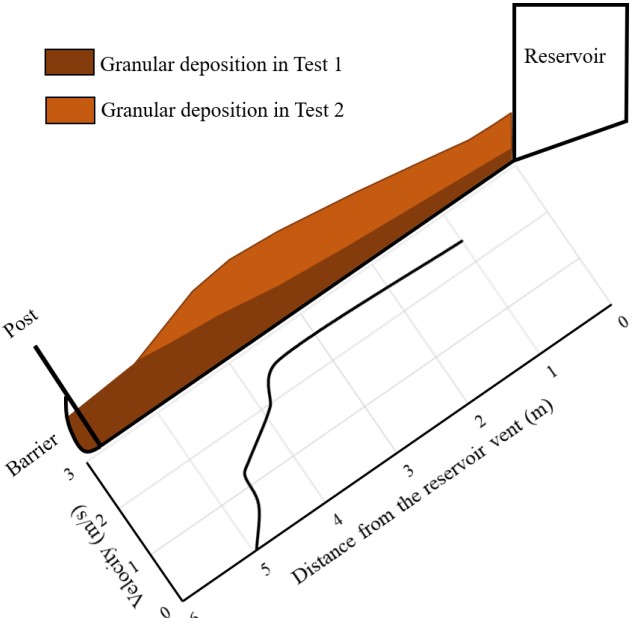


**Fig.10.** Side profile of the depositions in Test 1 and Test 2 and the velocity change of
the granular flow in Test 2 with the moving distance






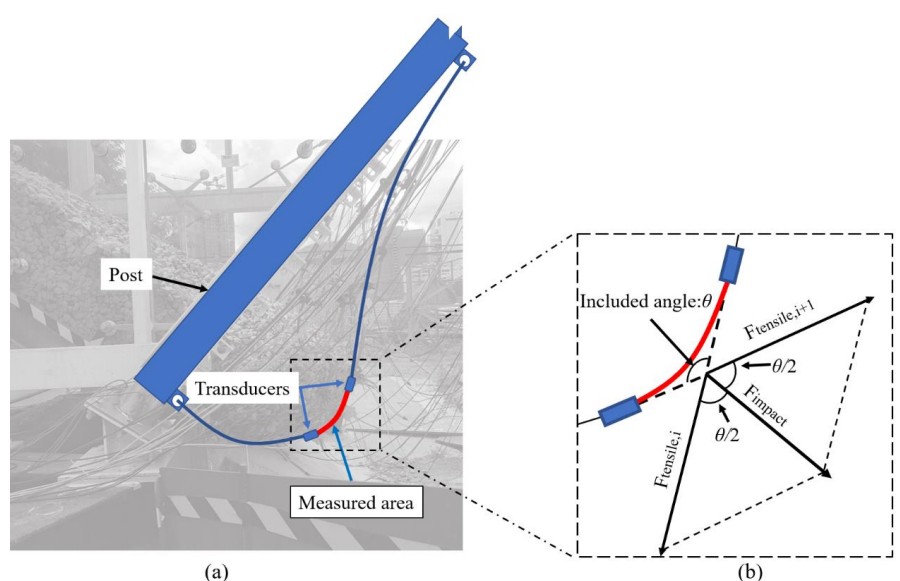


**Fig.11.** (a) sketch of the flexible barrier under the impact of a granular flow and (b)
the simplified force analysis of the measured area in the cross-section of
Transducer *i* and Transducer *i+1*






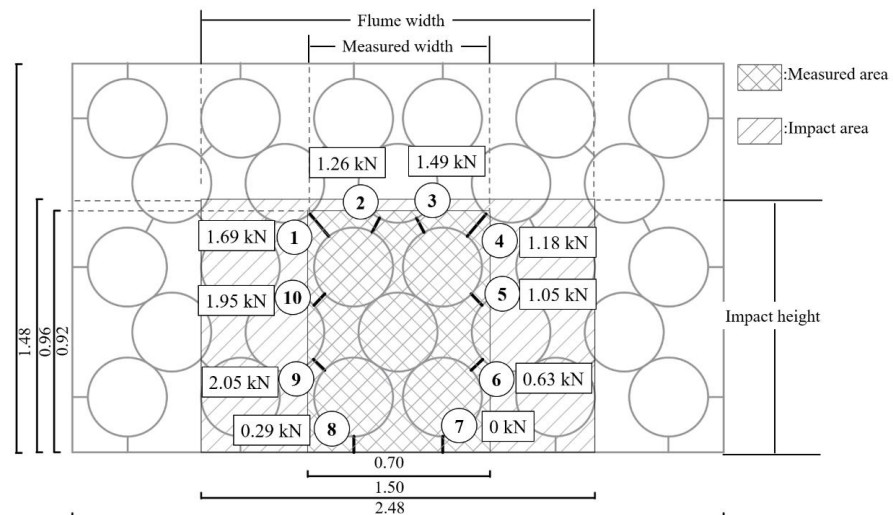



**Fig.12.** Sketch of the impact and measured area in Test 1 and the maximum tensile
forces measured from 10 mini tension link transducers under the impact of the
granular flow (unit in m)





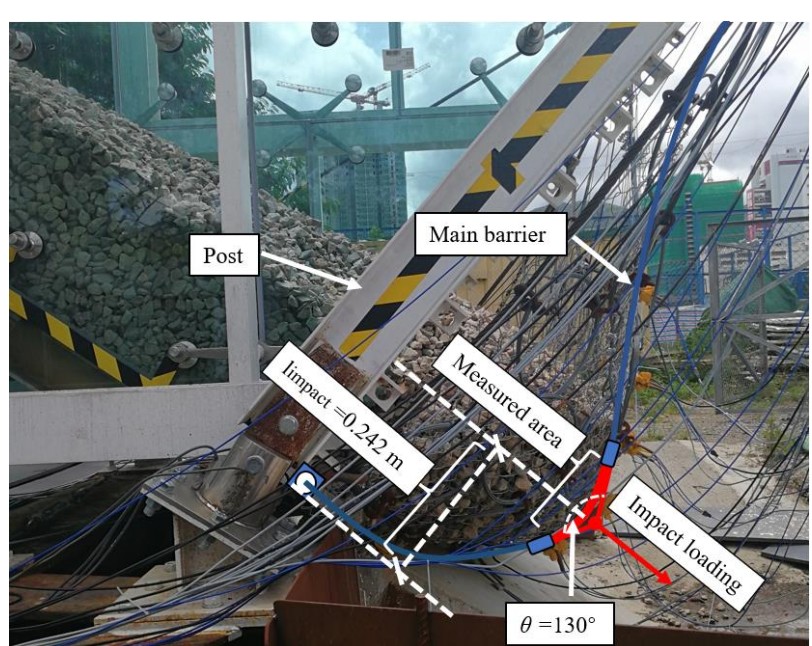


(a)

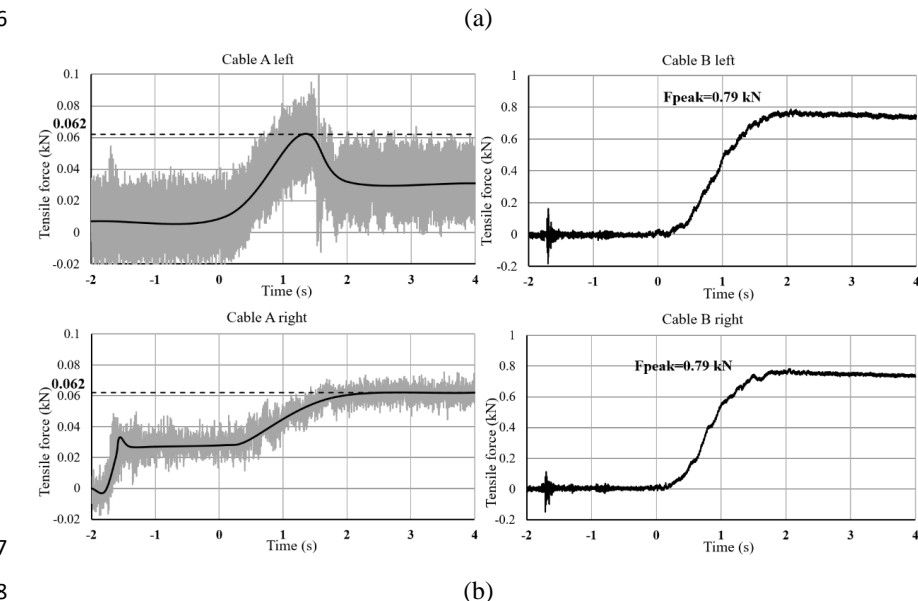


(b)

**Fig.13.** (a) photograph at the instant of the largest deformation with measured
parameters and (b) recorded forces and time by the tension link transducers
on the supporting cables in Test 1




(a)


(b)

**Fig.14.** (a) top-view and (b) left-side-view of sketches with the force analysis of the
posts and cables


