# Peer review of "Large Scale Physical Modelling Study of a Flexible Barrier under the Impact of Granular Flows"

_Natural Hazards and Earth System Sciences, 2018_

## Referee Comment (RC1) · Anonymous Referee #1 · 19 Jun 2018

The paper Large Scale Physical Modelling Study of a Flexible Barrier under the Impact of Granular Flows has addressed the open question how flexible barriers can be better designed to mitigate the impact loads from debris flows. The experimental devices and the experiments were very carefully designed. The large-scale granular flows tests were performed and the resultant forces measured on the flexible nets and the static structures were analyzed. Thus the authors reported that the flexible barrier can effectively reduce the granular impact loads. In addition, the traditional hydro-dynamic and hydro-static force estimation methods were validated using the acquired field data.

This study is solid and has great importance to practical applications in the field of natural hazards mitigation. The paper is very well written and the whole experiment is very clearly described. It is thus recommended to accept the paper for publication after

the following issues are clarified:

1) How did the authors define the word large-scale in their experiments? 2) In lines 195-197, how did the authors define the deposition height of the granular flow, and the maximum horizontal deformation of the flexible barrier? It is better to show them in the scratch. 3) What are the unique advantages of the experiments performed in this paper compared to the other researches, as the authors stated that an improved large-scale physical modelling facility for debris flow research has been conducted? 4) How many Test1 and Test2 experiments were performed by the authors? It would be great if the authors can comment how the experimental results vary between different rounds of experiments. 5) In Table 1, how did the authors determine the internal friction angle and the interface friction angle for granular flows? 6) In the 4th column of Table 3, the unit kN should not be italic.

---

## Referee Comment (RC2) · Anonymous Referee #2 · 20 Jun 2018

Page 5: value of 2.0 proposed by Wendeler in 2008: PHD Thesis ETH No 17916 Page 7: velocity of the flow only calculated by the high speed videos? Very roughly, no laser devices in front of the barrier? Page 8: 5 m/s can be for granular flow in the correct range but I am wondering about bulk density given with 1600 kg/m3 fitting not in the range of granular flow which normally have around 2000 kg/m3 (page 22) and more. Page 10: Second surge not realisitc for reality, because the material was already drained. How long was the time in between the two surges? In a real debris flow it happen all together very quickly, there is no time of drainage Page 12, line 279 it is Figure 12 instead of Figure 10. Page 16: Two tests is nothing for research background and statistic interpretation. You need more tests to interpret the results correctly. Second test is not useful because front was stopped, no dynamic impact

onto the barrier. Page 17: f explain and discuss the results together with table 3 page 24. It must be more clearly explained where the results come from. Page 17: I still believe that c=2.0 is representing the granular impact on flexible barriers but we need more test results.
* * *

---

## Author Comment (AC1) · 21 Jun 2018

1. How did the authors define the word large-scale in their experiments?

Reply: The definition of large-scale in our tests (Polyu model) is based on the definition of the large-scale physical model built by USGS (Iverson et al. 2010; Iverson 2015). The physical model built in Polyu site has similar dimensional parameters to the USGS debris-flow flume. Specifically, the capacity of testing material is 4.3 m3 in Polyu model compared to 10 m3 in USGS flume, and the width of the flume is 1.5 m in Polyu model compared to 2 m in USGS flume. Even though the length of the flume in Polyu model is much shorter than the length of USGS flume (7 m compared to 95 m), the flume in Polyu model is sufficient to generate granular flows with dynamic parameters similar to

real cases and debris flows in other large-scale tests. In the generated granular flow, the flow velocity (5 m/s), the measured impact force (10.96 kN) and the deposition mechanism are similar to the parameters of debris flows in literature (Bugnion and Wendeler 2010; Arattano and Marchi 2005). Thus, we regard Polyu model as a large-scale physical model.

2. In lines195-197, how did the authors define the deposition height of the granular flow, and the maximum horizontal deformation of the flexible barrier? It is better to show them in the scratch.

Reply: Thanks for the comments, we have added the definitions of the deposition height and the maximum horizontal deformation of the flexible barrier in Fig. 6 in the manuscript. A soft copy is attached in the following Fig. 1 in the reply.

3. What are the unique advantages of the experiments performed in this paper compared to the other researches, as the authors stated that an improved large-scale physical modelling facility for debris flow research has been conducted?

Reply: The description of the improved large-scale physical model used in our study is to emphasize that the physical modelling device is improved by a novel door opening system (see Page 6, Line 141-142). With the novel door opening system, the door can be flipped up quickly after triggering to minimize the interference from the door and increase the uniformity of the generated granular flows. Besides, a new measurement method is utilized to directly measure the impact forces on the flexible ring net (Section 4.1), which is another advantage of the experiments in this paper.

4. How many Test1 and Test2 experiments were performed by the authors? It would be great if the authors can comment how the experimental results vary between different rounds of experiments.

Reply: Thanks for the comments, we only did once for each test. We will consider conducting more tests in the future. However, it is difficult to perform more tests in a

short period due to the long preparation time of each test.

5. In Table 1, how did the authors determine the internal friction angle and the interface friction angle for granular flows?

Reply: The internal friction angle of the aggregates, which is regarded having the same value with the angle of repose (Hutter and Koch 1991), is measured by the pouring test introduced by Miura et al. (1997) and Zhou et al. (2014). The interface friction angle is determined by the tilting plane method introduced by Hutter and Koch (1991) and Zhou et al. (2014). The above description has been added in the manuscript (Page 7-8, Line 177-180).

6. In the 4th column of Table 3, the unit kN should not be italic.

Reply: Noted with thanks, we have corrected it in the manuscript.

References

Arattano, M. and Marchi, L., (2005). Measurements of debris flow velocity through cross-correlation of instrumentation data. Natural Hazards and Earth System Science, 5(1), 137-142.

Bugnion, L. and Wendeler, C., (2010). Shallow landslide full-scale experiments in combination with testing of a flexible barrier. WIT Transactions on Engineering Sciences, 67, 161-173.

Hutter, K. and Koch, T., (1991). Motion of a granular avalanche in an exponentially curved chute: experiments and theoretical predictions. Phil. Trans. R. Soc. Lond. A, 334(1633), 93-138.

Iverson, R.M., (2015). Scaling and design of landslide and debris-flow experiments. Geomorphology, 244, 9-20.

Iverson, R.M., Logan, M., LaHusen, R.G. and Berti, M., (2010). The perfect debris flow? Aggregated results from 28 large‐scale experiments. Journal of Geophysical

Research: Earth Surface, 115(F3).

Miura, K., Maeda, K. and Toki, S., (1997). Method of measurement for the angle of repose of sands. Soils and Foundations, 37(2), 89-96.

Zhou, G.G., Ng, C.W. and Sun, Q.C., (2014). A new theoretical method for analyzing confined dry granular flows. Landslides, 11(3), 369-384.

[Figure]

Side profiles figure with labels:
- Aggregates
- Deposition height
- Horizontal deformation

Legend:
- 0 s
- 0.51 s
- 0.71 s
- 0.91 s
- 1.21 s
- 1.61 s
- 2.01 s
- ∞

**Fig. 1.** Side profiles of deposited aggregates at different times in Test 1 (Fig.6)

---

## Author Response (AR1)

**"Large Scale Physical Modelling Study of a Flexible Barrier under the Impact of Granular Flows" (nhess-2018-131)**

**Reply to Review Comments from the Editor**

**by Dao-yuan TAN and Co-Authors**

The authors wish to thank the handling editor for his insightful and constructive comments on the manuscript and advice to us for improving the quality of the paper. The authors have taken full consideration of all those comments and made clarification and corrections in following tables:

| No. | Editor's comments | Reply |
|-----|-------------------|-------|
| 1 | Thank you for your response, please address the comments of the reviewers in an updated version. | Reply: Thank you for your reminder, all the comments of the reviewers and corresponding revisions have been addressed in the updated version in the attached files using black underlined fonts. |
| 2 | Very important that you address the issues of the Cd (drag coefficient) as I think most practicing engineers are interested in this value. Please note that a recent paper has been published in this field in the Canadian Geotechnical Journal: https://doi.org/10.1139/cgj-2016-0157. | Reply: This suggestion is very valuable and helpful to improve the draft: nhess-2018-131. The published paper by Wendeler *et al.* (2018) reviewed previous laboratory tests (Wendeler and Volkwein 2015) and full-scale field tests (Berger *et al.* 2011; Wendeler 2008) and proposed a stepwise load model to estimate the impact forces on the flexible barrier during the interaction with a debris flow. The hydro-dynamic approach and the hydro-static approach were applied in that model. The hydro-dynamic approach with the dynamic coefficient $c_w=2.0$ for granular flows suggested in that literature can accurately evaluate the impact forces on the flexible ring net measured in our large-scale tests. This literature also assumed that the impact loading from a debris flow applied on a flexible barrier is evenly distributed over the barrier. This assumption was proved by back-calculation using the data of field tests, which supports the assumption made in our draft that the impact pressure in the measured area can reflect the impact loading on the impact area of the flexible ring net (see Page 14 Lines 327-330).

To improve the quality of the draft: nhess-2018-131, the suggested literature has been comprehensively reviewed and appropriately cited. The revised and added contents are |

| | | marked as blue underlined fonts in the revised draft: |
| --- | --- | --- |
| | | 1) Page 3-4 (Lines 77-82). |
| | | 2) Page 4-5 (Lines 102-104). |
| | | 3) Review of the literature in Page 6 (Lines 132-144). |
| | | 4) Page 6 (Lines 147-148). |
| | | 5) Page 11 (Lines 267-272). |
| | | 6) The compared hydro-dynamic approaches with the test results used the dynamic coefficients proposed by Wendeler (2008): $c_w$=2.0 for granular flows and $c_w$=0.7 for debris flows with lower densities to verify their application in predicting impact forces of dry granular flows. See Page 17 (Lines 405-407), Page 18 (Lines 412-423) and Table 3 (Page 27). |
| | | 7) Add the suggested paper into the reference list: Page 24 (Lines 576-578). |

**References:**

Berger, C., McArdell, B.W., Schlunegger, F. (2011). Direct measurement of channel erosion by debrisflows, Illgraben, Switzerland. J. Geophys. Res. 116, F01002, doi: 10.1029/2010JF001722: 18 p.

Wendeler, C. S. I. (2008). Murgangrückhalt in Wildbächen. Grundlagen zu Planung und Berechnung von flexiblen Barrieren. ETH.

Wendeler, C., Volkwein, A., McArdell, B.W. and Bartelt, P. (2018). Load model for designing flexible steel barriers for debris flow mitigation. Canadian Geotechnical Journal, (ja).

Wendeler, C., and Volkwein, A. (2015). Laboratory tests for the optimization of mesh size for flexible debris-flow barriers. Natural Hazards and Earth System Sciences, 15(12).

**"Large Scale Physical Modelling Study of a Flexible Barrier under the Impact of Granular Flows" (nhess-2018-131)**

**Reply to Review Comments from the Referees**

**by Dao-yuan TAN and Co-Authors**

The authors wish to thank the referees for their insightful and constructive comments on the manuscript and advice to us for improving the quality of the paper. The authors have taken full consideration of all those comments and made clarification and corrections in following tables:

| No. | Referee 1's comments | Reply |
|---|---|---|
| 1 | How did the authors define the word large-scale in their experiments? | Reply: This is a very good question. The definition of large-scale in our tests (PolyU model) is based on the definition of the large-scale physical model built by USGS (Iverson *et al.* 2010; Iverson 2015). The physical model built in PolyU site has similar dimensional parameters to the USGS debris-flow flume. Specifically, the capacity of testing material is 5 $m^3$ in PolyU model compared to 10 $m^3$ in USGS flume, and the width of the flume is 1.5 m in PolyU model compared to 2 m in USGS flume. Even though the length of the flume in PolyU model is much shorter than the length of USGS flume (7 m compared to 95 m), the flume in PolyU model is sufficient to generate debris flows with dynamic parameters similar to real cases. In the trial tests, the generated watery flood can reach a velocity higher than 8 m/s during the flowing down. In the generated granular flow, the flow velocity (5 m/s), the measured impact force (10.96 kN) and the deposition mechanism are similar to the parameters of debris flows in literatures (Bugnion and Wendeler 2010; Arattano and Marchi 2005). Thus, we regard Polyu model as a large-scale physical model. Related explanation has been added into the manuscript in Page 8 (Lines 179-189). |

| 2 | In lines195-197, how did the authors define the deposition height of the granular flow, and the maximum horizontal deformation of the flexible barrier? It is better to show them in the scratch. | Reply: Thanks for the valuable comment, we have added the definitions of the deposition height and the maximum horizontal deformation of the flexible barrier in Page 10, Lines 239-242 and Fig.5 in Page 33. |
|---|---|---|
| 3 | What are the unique advantages of the experiments performed in this paper compared to the other researches, as the authors stated that an improved large-scale physical modelling facility for debris flow research has been conducted? | Reply: The description of the improved large-scale physical model is to emphasize that the physical modelling device is improved by a fast door opening system (see Page 7, Line 168). With the fast door opening system, the door can be flipped up quickly (shorter than 0.5 s) after triggering to minimize the interference from the door and increase the uniformity of the generated granular flows. Besides, a new method is utilized to directly measure the impact forces on the flexible ring net (Section 4.1), which is another advantage of the experiment device in this paper. |
| 4 | How many Test1 and Test2 experiments were performed by the authors? It would be great if the authors can comment how the experimental results vary between different rounds of experiments. | Reply: Thanks for the comments, and we only did once for each test. We will consider conducting more tests in the future by changing parameters of granular flows and flexible barriers. However, it is difficult to perform more tests within a short period due to the long preparation time of each test. |
| 5 | In Table 1, how did the authors determine the internal friction angle and the interface friction angle for granular flows? | Reply: The internal friction angle of the aggregates, which is regarded having the same value with the angle of repose (Hutter and Koch 1991), is measured by the pouring test introduced by Miura *et al.* (1997) and Zhou *et al.* (2014). The interface friction angle is determined by the tilting plane method introduced by Hutter and Koch (1991) and Zhou *et al.* (2014). The above description has been added in the manuscript (Page 9, Lines 217-221). |
| 6 | In the 4th column of Table 3, the unit kN should not be italic. | Reply: Noted with thanks, we have corrected it in the manuscript. |

| No. | Referee 2's comments | Reply |
|-----|----------------------|-------|
| 1 | Page 5: value of 2.0 proposed by Wendeler in 2008: PHD Thesis ETH No 17916 | Reply: Thanks for your correction, we have corrected this citation error in Page 5 (Line 122-124) and Table 3. |
| 2 | Page 7: velocity of the flow only calculated by the high speed videos? Very roughly, no laser devices in front of the barrier? | Reply: The velocity of the granular flow was measured from continuous photographs taken by the side-view high-speed camera. To reduce the measuring error, the impact velocity of the granular flow is calculated from the average value of the velocities of 5 particles measured from 5 continuous photographs before the impact with the assistance of the reference lines attached to the flume. Related explanation has been added into the manuscript in Page 9 (Lines 209-213). We agree that more measuring devices will increase the accuracy of measurement. |
| 3 | Page 8: 5 m/s can be for granular flow in the correct range but I am wondering about bulk density given with 1600 kg/m3 fitting not in the range of granular flow which normally have around 2000 kg/m3 (page 22) and more. | Reply: We agree that the typical bulk density of granular flows is around 2000 kg/m$^3$, but the testing material in our study is dry aggregate, which has a lower bulk density. |
| 4 | Page 10: Second surge not realisitc for reality, because the material was already drained. How long was the time in between the two surges? In a real debris flow it happen all together very quickly, there is no time of drainage | Reply: The time interval between two tests is around 2 weeks, because we need at least 2 weeks to prepare a test. We agree that the drainage of the debris deposition should be considered in the study of multiple debris flows. In our study, the research subject is dry granular flow. Thus, drainage should not be a problem. |
| 5 | Page 12, line 279 it is Figure 12 instead of Figure 10. | Reply: Thanks for your correction, we have corrected it in the manuscript. |

| 6 | Page 16: Two tests is nothing for research background and statistic interpretation. You need more tests to interprete the results correctly. Second test is not useful because front was stopped, no dynamic impact onto the barrier. | Reply: We agree that more tests can enhance the reliability of the quantitative conclusions drawn in this study, but it is difficult to perform more tests in a short period due to the long preparation time of a large-scale test. The granular flow in Test 2 was stopped before it can reach the flexible barrier due to the poor fluidity of dry granular flows, but it still can provide valuable data in the study of the motion and the deposition of the second surge in a multiple granular flow event. |
|---|---|---|
| 7 | Page 17: explain and discuss the results together with table 3 page 24. It must be more clearly explained where the results come from. | Reply: Thanks for the valuable comments. With the conclusions drawn from Table 3, it can be preliminarily concluded that the impact force on the flexible ring net and on the supporting structures should be estimated separately using different simple approaches. Thus, the design of a flexible barrier for debris flow mitigation can be optimized by dimensioning and designing the flexible ring net and the supporting structures individually with appropriate design loadings, which provides a safer and more economical design method. A specified explanation has been added into the manuscript (Lines 474 to 479). We have also corrected the citation error of the hydro-dynamic approach with the dynamic coefficient of 2.0 in Table 3. |
| 8 | Page 17: I still believe that c=2.0 is representing the granular impact on flexible barriers but we need more test results. | Reply: We agree that the hydro-dynamic approach with the dynamic coefficient of 2.0 can correctly represent the impact of a granular flow on the flexible barrier based on the comparisons in our study. More tests are under consideration to further verify the coefficients in simple approaches using different debris material such as muddy debris flows. |

[revised manuscript text omitted]

---

## Author Response (AR2)

**"Large Scale Physical Modelling Study of a Flexible Barrier under the Impact of Granular Flows" (nhess-2018-131)**

**Reply to Review Comments from the Editor**

**by Dao-yuan TAN and Co-Authors**

The authors wish to thank the handling editor for his insightful and constructive comments on the manuscript and advice to us for improving the quality of the draft paper. The authors have taken full consideration of all those comments and made clarification and corrections in following tables:

| No. | Editor's comments | Reply |
|---|---|---|
| 1 | Thank you for your response, please address the comments of the reviewers in an updated version. | Reply: Thank you for your reminder, all the comments of the reviewers and corresponding revisions have been addressed in the updated version in the attached files using black underlined fonts. |
| 2 | Very important that you address the issues of the Cd (drag coefficient) as I think most practicing engineers are interested in this value. Please note that a recent paper has been published in this field in the Canadian Geotechnical Journal: https://doi.org/10.1139/cgj-2016-0157. | Reply: This suggestion is very valuable and helpful to improve the draft: nhess-2018-131. The published paper by Wendeler *et al.* (2018) reviewed previous laboratory tests (Wendeler and Volkwein 2015) and full-scale field tests (Berger *et al.* 2011; Wendeler 2008) and proposed a stepwise load model to estimate the impact forces on the flexible barrier during the interaction with a debris flow. The hydro-dynamic approach and the hydro-static approach were applied in that model. The hydro-dynamic approach with the dynamic coefficient $c_w$=2.0 for granular flows suggested in that literature can accurately evaluate the impact forces on the flexible ring net measured in our large-scale tests. This literature also assumed that the impact loading from a debris flow applied on a flexible barrier is evenly distributed over the barrier. This assumption was proved by back-calculation using the data from field tests, which supports the assumption made in our draft paper that the impact pressure in the measured area can reflect the impact loading on the impact area of the flexible ring net (see Page 14 Lines 332-335).

To improve the quality of the draft paper: nhess-2018-131, the suggested literature has been comprehensively reviewed and appropriately cited. The revised and added |

| | | contents are marked as blue underlined fonts in the revised draft:
1) Page 3-4 (Lines 77-82).
2) Page 4-5 (Lines 102-104).
3) Review of the literature in Page 6 (Lines 132-144).
4) Page 6 (Lines 147-148).
5) Pages 11-12 (Lines 272-277).
6) We used the measured impact forces of the dry granular flow to verify the hydro-dynamic approaches with two dynamic coefficients proposed by Wendeler (2008): $c_w$=2.0 for granular flows and $c_w$=0.7 for muddy debris flows. See Pages 17-18 (Lines 410-412), Page 18 (Lines 417-428) and Table 3 (Page 27).
7) Add the suggested literature into the reference list: Page 24 (Lines 585-587). |
|---|---|---|
| 3 | I have one final question: did you measure the "pile-up" density behind the net? In the table you report the bulk density (1600 kg/m3), but I was wondering if you have any idea of the compacted density behind the net. | Reply: Thanks for raising this valuable question and the insightful suggestion. We did not measure the compacted density behind the net in the conducted tests. But we believe that both the flowing density before the impact and the density during the deposition after impact shall be measured in the future tests with the assistance of laser devices and force plates. We had measured only the loose dry bulk density of the aggregate material according to ASTM C29/C29M-91a (ASTM 2009) before we conducted the dry granular flow impact tests. The bulk density of the dry aggregate with the value of 1600 kg/m$^3$ fits well with the testing results in the literatures (Rücknagel *et al.* 2007, Raj *et al.* 2014, Yahia and Kabagire 2014). To avoid the confusion, we have clarified the determination of the dry bulk density in Page 9 (Lines 220-221) and Page 22 (Lines 511-512) for adding the reference.

When a dry granular flow is stopped by a flexible barrier, the deposited aggregate is compacted by the oncoming debris front, and the bulk density of the aggregate deposited behind the barrier increases correspondingly. Therefore, the density used in the hydro-static model should be different from the density used in the hydro-dynamic model. However, Wendeler *et al.* (2018) used the same debris density in the hydro-static model as that in the |

| | | hydro-dynamic model, because the magnitude of the static pressure could be reduced due to the stability increase of the deposited debris compared to the flowing debris. Undoubtedly, it is highly worthy to quantify the density and the stability change before and after the impact and the deposition process in the future studies. |
|---|---|---|

**Reply to Review Comments from the Referees**

**by Dao-yuan TAN and Co-Authors**

The authors wish to thank the referees for their insightful and constructive comments on the manuscript and advice to us for improving the quality of the draft paper. The authors have taken full consideration of all those comments and made clarification and corrections in following tables:

**Replies to Referee 1's comments:**

| No. | Referee 1's comments | Reply |
|---|---|---|
| 1 | How did the authors define the word large-scale in their experiments? | Reply: This is a very good question. The definition of large-scale in our tests (PolyU model) is based on the definition of the large-scale physical model built by USGS (Iverson *et al.* 2010; Iverson 2015). The physical model built in PolyU site has similar dimensional parameters to the USGS debris-flow flume. Specifically, the capacity of testing material is 5 $m^3$ in PolyU model compared to 10 $m^3$ in USGS flume, and the width of the flume is 1.5 m in PolyU model compared to 2 m in USGS flume. Even though the length of the flume in PolyU model is much shorter than the length of USGS flume (7 m compared to 95 m), the flume in PolyU model is sufficient to generate debris flows with dynamic parameters similar to real cases. In the trial tests, the generated watery flood can reach a velocity higher than 8 m/s during the flowing down. In the generated granular flow, the flow velocity (5 m/s), the measured impact force (10.96 kN) and the deposition mechanism are similar to the parameters of debris flows in literatures (Bugnion and Wendeler 2010; Arattano and Marchi 2005). Thus, we regard Polyu model as a large-scale physical model. Related explanation has been added into the manuscript in Page 8 (Lines 179-189). |

| 2 | In lines195-197, how did the authors define the deposition height of the granular flow, and the maximum horizontal deformation of the flexible barrier? It is better to show them in the scratch. | Reply: Thanks for the valuable comment, we have added the definitions of the deposition height and the maximum horizontal deformation of the flexible barrier in Page 10, Lines 244-247 and Fig.5 in Page 34. |
|---|---|---|
| 3 | What are the unique advantages of the experiments performed in this paper compared to the other researches, as the authors stated that an improved large-scale physical modelling facility for debris flow research has been conducted? | Reply: The description of the improved large-scale physical model is to emphasize that the physical modelling device is improved by a fast door opening system (see Page 7, Line 168). With the fast door opening system, the door can be flipped up quickly (shorter than 0.5 s) after triggering to minimize the interference from the door and increase the uniformity of the generated granular flows. Besides, a new method is utilized to directly measure the impact forces on the flexible ring net (Section 4.1), which is another improvement of the experiment device in this paper. |
| 4 | How many Test1 and Test2 experiments were performed by the authors? It would be great if the authors can comment how the experimental results vary between different rounds of experiments. | Reply: Thanks for the comments, and we only did once for each test. We will consider conducting more tests in the future by changing parameters of granular flows and flexible barriers. However, it is difficult to perform more tests within a short period due to the long preparation time of each test. |
| 5 | In Table 1, how did the authors determine the internal friction angle and the interface friction angle for granular flows? | Reply: The internal friction angle of the aggregate, which is regarded having the same value with the angle of repose (Hutter and Koch 1991), is measured by the pouring test introduced by Miura *et al.* (1997) and Zhou *et al.* (2014). The interface friction angle is determined by the tilting plane method introduced by Hutter and Koch (1991) and Zhou *et al.* (2014). The above description has been added in the manuscript (Page 9, Lines 221-225). |
| 6 | In the 4th column of Table 3, the unit kN should not be italic. | Reply: Noted with thanks, we have corrected it in the manuscript. |

**Replies to Referee 2's comments:**

| No. | Referee 2's comments | Reply |
|-----|----------------------|-------|
| 1 | Page 5: value of 2.0 proposed by Wendeler in 2008: PHD Thesis ETH No 17916 | Reply: Thanks for your correction, we have corrected this citation error in Page 5 (Line 122-124) and Table 3. |
| 2 | Page 7: velocity of the flow only calculated by the high speed videos? Very roughly, no laser devices in front of the barrier? | Reply: Thanks for your valuable suggestions. We agree that more measuring devices will increase the accuracy of measurement, and we will consider adding more measuring devices such as laser devices to better determine the flow velocity and depth. However, in this study, the velocity of the granular flow was merely measured from continuous photographs taken by the side-view high-speed camera. To increase the accuracy of the measurement, two actions were taken: firstly, we set the location and the shooting angle of the side-view high speed camera very carefully to make sure that the camera was perpendicular to the transparent side wall of the flume; secondly, the impact velocity of the granular flow was determined from the average value of the velocities of 5 particles measured from 5 continuous photographs with the assistance of the reference lines attached to the flume. Related explanation of the measurement has been added into the manuscript in Page 9 (Lines 209-216). |
| 3 | Page 8: 5 m/s can be for granular flow in the correct range but I am wondering about bulk density given with 1600 kg/m3 fitting not in the range of granular flow which normally have around 2000 kg/m3 (page 22) and more. | Reply: We agree that the typical bulk density of granular debris flows is around 2000 kg/m$^3$, but the testing material in our study is dry aggregate with a large percentage of void space and a lower bulk density. We measured the loose dry bulk density of the aggregate material according to ASTM C29/C29M-91a (ASTM 2009) before we conducted the dry granular flow impact tests. The bulk density of the dry aggregate with the value of 1600 kg/m$^3$ fits well with the testing results in the literatures (Rücknagel *et al.* 2007, Raj *et al.* 2014, Yahia and Kabagire 2014). To avoid the confusion, we have clarified the determination of the dry bulk density in Page 9 (Lines 220-221) and Page 22 (Lines 511-512) for adding the reference. In the future, we will conduct more tests using a mixture of aggregate and |

| | | saturated slurry, which is predicted to have higher densities much closer to the common value of 2000 kg/m$^3$. |
|---|---|---|
| 4 | Page 10: Second surge not realisitc for reality, because the material was already drained. How long was the time in between the two surges? In a real debris flow it happen all together very quickly, there is no time of drainage | Reply: The time interval between two tests is around 2 weeks, because we need at least 2 weeks to prepare a test. We agree that the drainage of the debris deposition should be considered in the study of multiple debris flows, and we plan to do related tests in the future. In our study, the research subject is dry granular flow. Thus, drainage should not be a problem. Many thanks for your suggestions, and we will be more careful in conducting impact tests with saturated debris flows in the future. |
| 5 | Page 12, line 279 it is Figure 12 instead of Figure 10. | Reply: Thanks for your correction, we have corrected it in the manuscript. |
| 6 | Page 16: Two tests is nothing for research background and statistic interpretation. You need more tests to interpret the results correctly. Second test is not useful because front was stopped, no dynamic impact onto the barrier. | Reply: Thanks for your suggestions. We strongly agree that more tests can enhance the reliability of the quantitative conclusions drawn in this study, but one successful large-scale physical modelling test can clearly investigate the impact mechanisms of a granular flow with poor fluidity on a flexible barrier. We also believe that the verification of simple approaches using the impact forces on different components is also valuable to the future research and the design of debris flow flexible barrier. Besides, the verification results fit well with the conclusions drawn in the literatures (Wendeler 2008, Wendeler *et al.* 2018). Even the granular flow in Test 2 was stopped before it can reach the flexible barrier due to the poor fluidity of dry granular flows, it still can provide valuable data and reference to the study of the motion and the deposition of the second surge in a multiple debris flow event. |
| 7 | Page 17: explain and discuss the results together with table 3 page 24. It must be more clearly explained where the results come from. | Reply: Thanks for the valuable suggestions. With the conclusions drawn from the large-scale tests presented in this draft paper, it can be preliminarily concluded that the impact force on the flexible ring net and on the supporting structures are different due to the large deformation of the flexible ring net, thus the loadings on them should be estimated |

| | | separately using different simple approaches or an appropriate value of Loading Reduction Rate (LRR). Thus, the design of a flexible barrier for debris flow mitigation can be optimized by dimensioning and designing the flexible ring net and the supporting structures individually with appropriate design loadings, which provides a safer and more economical design method.

 A specified explanation has been added into the manuscript (Lines 477 to 485). |
|---|---|---|
| 8 | Page 17: I still believe that c=2.0 is representing the granular impact on flexible barriers but we need more test results. | Reply: We agree. The dynamic coefficient of 2.0 has been verified by the measured impact force on the flexible ring net in the large-scale test (see Table 3). This approach can accurately estimate the impact of a granular flow on a flexible barrier. More tests are under consideration to further verify simple approaches using different debris materials such as muddy debris flows. |

[revised manuscript text omitted]